# Structural insight into the dual function of LbpB in mediating Neisserial pathogenesis

**Ravi Yadav[1,2], Srinivas Govindan[3], Courtney Daczkowski[4], Andrew Mesecar[2,4], Srinivas Chakravarthy[5], Nicholas Noinaj[2,6]***

[1]Purdue University Interdisciplinary Life Sciences Program, West Lafayette, United States; [2]Department of Biological Sciences,Purdue University, West Lafayette, United States; [3]Weldon School of BiomedicalEngineering, Purdue University, West Lafayette, United States; [4]Department of Biochemistry, Purdue University, West Lafayette, United States; [5]BIO-CAT, Illinois12 Institute of Technology, Lemont, United States; [6]Purdue Institute for Inflammation, Immunology and Infectious Disease, Purdue University, West Lafayette, United States

**Abstract** Lactoferrin-binding protein B (LbpB) is a lipoprotein present on the surface of *Neisseria* that has been postulated to serve dual functions during pathogenesis in both iron acquisition from lactoferrin (Lf), and in providing protection against the cationic antimicrobial peptide lactoferricin (Lfcn). While previous studies support a dual role for LbpB, exactly how these ligands interact with LbpB has remained unknown. Here, we present the structures of LbpB from *N. meningitidis* and *N. gonorrhoeae* in complex with human holo-Lf, forming a 1:1 complex and confirmed by size-exclusion chromatography small-angle X-ray scattering. LbpB consists of N- and C-lobes with the N-lobe interacting extensively with the C-lobe of Lf. Our structures provide insight into LbpB's preference towards holo-Lf, and our mutagenesis and binding studies show that Lf and Lfcn bind independently. Our studies provide the molecular details for how LbpB serves to capture and preserve Lf in an iron-bound state for delivery to the membrane transporter LbpA for iron piracy, and as an antimicrobial peptide sink to evade host immune defenses.

*For correspondence:
nnoinaj@purdue.edu

**Competing interest:** The authors declare that no competing interests exist.

## Introduction

There are 17 species of *Neisseria* that colonize humans with only two being pathogenic, *N. meningitidis* and *N. gonorrhoeae* (*Seifert, 2019*; *Tone Tønjum, 2017*). *N. meningitidis* asymptomatically colonizes ~10 % of the world's population, however, once pathogenic can lead to meningitidis and sepsis and lead to high fatality rates in the absence of immediate treatment (*Caugant and Brynildsrud, 2020*; *Read, 2019*; *Siddiqui et al., 2021*). *N. gonorrhoeae* causes the sexually transmitted disease gonorrhoea (*Boyajian et al., 2016*). The Centers for Disease Control and Prevention (CDC) have categorized *N. gonorrhoeae* as an urgent threat to public health, which requires immediate and aggressive actions to combat the emerging antibiotic-resistant strains recently discovered (*Aitolo et al., 2021*; *Prevention (CDC), 2011*). Although vaccines are now available against all pathogenic serogroups of *N. meningitidis*, there are well-documented drawbacks and limitations of these vaccines (*Deasy and Read, 2011*; *Pizza et al., 2020*). And despite decades of research, there is still no vaccine against gonococcal infections. Therefore, there is an urgent need for the development of new and improved therapeutics for protection against these human pathogens, particularly against *N. gonorrhoeae*.

Acquisition of nutrients is an essential step during Neisserial pathogenesis (*Stork et al., 2013*; *Zackular et al., 2015*). Iron is an essential nutrient for survival, growth, and virulence (*Grifantini*

*et al., 2003*). Given that free iron is essentially non-existent in the human host, *Neisseria* have evolved specialized surface receptors to hijack iron from host iron-binding proteins such as transferrin (Tf) and lactoferrin (Lf) (*Lee and Schryvers, 1988*; *Schryvers and Lee, 1989*; *Schryvers and Morris, 1988*; *Yadav et al., 2019*). One of the more well-studied metal acquisition systems in *Neisseria* is the transferrin-binding protein (Tbp) system, which consists of an a TonB-dependent transporter called TbpA and a lipoprotein co-receptor called TbpB (*Cornelissen et al., 1992*; *Irwin et al., 1993*; *Noinaj et al., 2012a*; *Noinaj et al., 2012b*; *Noinaj et al., 2010*; *Pintor et al., 1998*). Here, TbpB serves to capture and deliver iron-loaded Tf to TbpA for iron extraction and import across the outer membrane in a process that requires energy from the Ton complex at the inner membrane (*Cornelissen et al., 1992*; *Cornelissen and Hollander, 2011*; *Noinaj et al., 2010*).

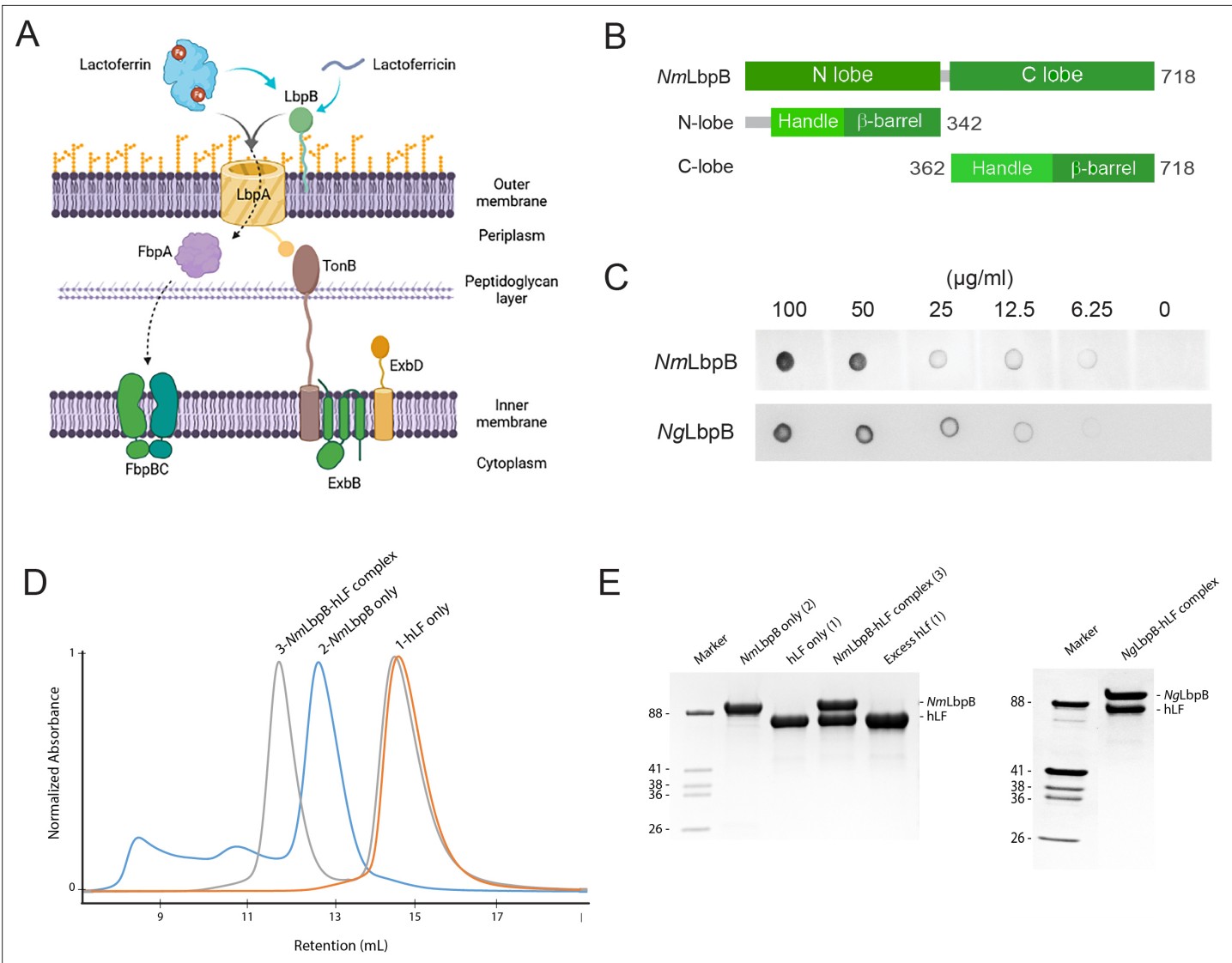

**Figure 1.** Lactoferrin-binding protein B (LbpB) forms a stable 1:1 complex with lactoferrin. (**A**) The proposed role of Lbp system in iron acquisition from lactoferrin and protection from lactoferricin (Biorender). (**B**) Summary of LbpB constructs used in this study. (**C**) Solid-phase-binding assay of holo-lactoferrin (Lf) binding to *N. meningitidis* LbpB (*Nm*LbpB; anti-Lf) and *N. gonorrhoeae* LbpB (*Ng*LbpB; Lf-horse radish peroxidase [HRP]). (**D**) Formation of the *Nm*LbpB–Lf complex over size-exclusion chromatography (SEC) from purified components. A leftward shift is observed for the complex compared to the individual components indicating the formation of the complex. (**E**) Sodium dodecyl sulphate-polyacrylamide gel electrophoresis (SDS–PAGE) analysis of the *Nm*LbpB–Lf complex formed from panel D, indicating the formation of the complex at a 1:1 ratio (lane 4). Similarly, the *Ng*LbpB–Lf complex was formed by SEC from purified components which also formed a 1:1 complex as shown by SDS–PAGE analysis.

The online version of this article includes the following source data for figure 1:

**Source data 1.** LbpB forms a stable 1:1 complex with lactoferrin.

A second analogous system is called the lactoferrin-binding protein (Lbp) system, which is composed of LbpA and LbpB, targets Lf as an iron source (*Figure 1A*; *Biswas and Sparling, 1995*; *Gray-Owen and Schryvers, 1996*; *Lee and Schryvers, 1988*; *Noinaj et al., 2013*; *Schryvers et al., 1998*; *Schryvers and Morris, 1988*). However, less is known structurally and functionally about the Lbp system as no structures of Lbp proteins in complex with Lf have been reported. While all strains of *N. meningitidis* contain a functional Lbp system, it is only observed in roughly half of the *N. gonorrhoeae* strains (*Anderson et al., 2003*; *Biswas et al., 1999*). Studies hint that the Lbp system may serve as a backup option to the Tbp system to assist in mediating pathogenesis within diverse host environments, as a Neisserial strain lacking the Tbp proteins were still able to use the Lbp proteins to cause infection (*Anderson et al., 2003*). Further, it was also shown that a strain expressing both systems was more competitive than a strain expressing just the Tbp system alone (*Anderson et al., 2003*). Moreover, anti-sera raised against the Lbp proteins displays bactericidal activity, exemplifying the promise of this system as a target for therapeutic intervention (*Pettersson et al., 2006*).

Given its sequence homology to TbpB, LbpB is predicted to have a similar domain organization consisting of an N- and C-lobes, with each lobe composed of an eight-stranded β-barrel sitting adjacent to a handle domain (*Beddek and Schryvers, 2010*; *Brooks et al., 2014*; *Ekins et al., 2004*; *Lee and Schryvers, 1988*; *Noinaj et al., 2013*; *Ostan et al., 2017*; *Schryvers and Lee, 1989*). This was partially confirmed when the structure of the N-lobe of LbpB was reported, showing conservation with the N-lobe of TbpB (*Brooks et al., 2014*). However, reports have varied on the mode of interaction between LbpB and Lf. Docking studies predicted that N-lobe of Lf interacts with LbpB (*Arutyunova et al., 2012*; *Brooks et al., 2014*), while hydrogen-deuterium exchange studies, combined with site-directed mutagenesis, showed that the C-lobe of Lf interacts with LbpB (*Ostan et al., 2017*). Homology modelling studies predicted that binding would align with that observed in TbpB–Tf (*Noinaj et al., 2013*), however, no structure of LbpB in complex with Lf has been reported.

In addition to its role in iron acquisition, LbpB has been shown to provide protection against the host cationic antimicrobial peptide lactoferricin (Lfcn), an 11-residue metabolized fragment from the N-lobe of Lf (*Figure 1A*; *Morgenthau et al., 2014a*; *Morgenthau et al., 2012*; *Morgenthau et al., 2014b*). Studies have been conflicting on exactly where LbpB interacts with Lfcn to neutralize this threat. One model proposes that it binds along the N-lobe (*Brooks et al., 2014*), while another model points to several anionic regions along the C-lobe (*Morgenthau et al., 2014a*; *Morgenthau et al., 2014b*). In addition to protecting the surface of *Neisseria* directly, LbpB has been shown to also be proteolytically cleaved by NalP, which enables it to diffuse into the host environment to pre-emptively neutralize Lfcn (*Roussel-Jazédé et al., 2010*). This additional role for LbpB serving as an antimicrobial peptide sink during Neisserial pathogenesis, in conjunction with its absence in many Neisserial strains despite LbpA being present, has cast some doubt onto its importance in iron import (*Morgenthau et al., 2014a*).

To further characterize the role of LbpB in Neisserial pathogenesis, here we report the crystal structure of *N. meningitidis* LbpB (*Nm*LbpB) in complex with Lf and the cryoEM structure of *N. gonorrhoeae* LbpB (*Ng*LbpB) in complex with Lf. Our structures were further confirmed by size-exclusion chromatography small-angle X-ray scattering (SEC-SAXS) studies which also indicated that Lf and Lfcn have non-overlapping binding sites on LbpB with a 1:1 binding ratio for each. Structure-guided mutagenesis studies then probed the binding interfaces between *Nm*LbpB and both Lf and Lfcn using biophysical and biochemical techniques. Our results support the model where LbpB serves dual functions, both in iron piracy from holo-Lf and as an antimicrobial peptide sink, during Neisserial pathogenesis.

## Results

### Complex formation of Neisserial LbpB with human Lf

For complex formation and structural studies, we recombinantly expressed and purified the soluble construct of LbpB from *N. meningitidis* (*Nm*LbpB) (residues 20–737) and *N. gonorrhoeae* (*Ng*LbpB) (residues 20–728) (*Figure 1B*). Solid-phase-binding assays showed that these purified LbpB constructs were sufficient for binding holo-Lf (*Figure 1C* and *Figure 1—source data 1*). For large-scale studies, Lf was purchased and further purified using SEC. Complexes of LbpB with Lf were formed upon incubation of purified LbpB with an excess of Lf and separated using SEC, with the complex eluting as

a left-shifted peak (*Figure 1D*). Sodium dodecyl sulphate-polyacrylamide gel electrophoresis (SDS–PAGE) analysis confirmed the peak contained both components and suggested a 1:1 stoichiometric complex for both the *Nm*LbpB and *Ng*LbpB complexes with Lf (*Figure 1E* and *Figure 1—source data 1*).

## Solution studies of LbpB complexes with Lf

It has been previously reported that *Nm*LbpB may have multiple Lf-binding sites and may form higher-order complexes (*Ostan et al., 2017*). However, in our studies with complex formation by SEC, we observed that *Nm*LbpB appears to form a stable 1:1 complex with Lf. To further confirm this, we studied the stoichiometry of LbpB (*Nm* and *Ng*) in complex with Lf in solution using inline SEC-SAXS. The SAXS curve provides information about the shape, molecular weight, and maximum length of a protein or protein complex. It is a great tool for comparing the structural effects of different ligands bound LbpB and for comparing the structures of LbpB from *Nm* and *Ng* to determine how closely they match. Further, SAXS allows us to compare how closely our structures match what is observed in solution by seeing how well the simulated SAXS curve from our structure fits the experimental SAXS curve. For our experiments here, our SEC-SAXS studies showed that the individual LbpB and Lf proteins existed as monomers in solution, while the *Nm*LbpB–Lf complex had an experimental molecular weight of 159.6 kDa corresponding to 1:1 stoichiometry complex of *Nm*LbpB and Lf (calculated molecular weight is 156.6 kDa) (*Figure 2A and B* and *Supplementary file 1a*). The calculated radius of gyration ($R_g$) and maximal intramolecular dimension ($D_{max}$) are both consistent with a 1:1 stoichiometric complex. Further, we used SEC-SAXS analysis to show that in solution, *Ng*LbpB alone and a complex with Lf match closely to those with *Nm*LbpB (*Figure 2A and B*, and SI *Supplementary file 1*).

## The crystal structure of *Nm*LbpB in complex with human Lf

Static small-angle X-ray scattering (SAXS) analysis showed that *Nm*LbpB alone tends to aggregate in solution, while the *Nm*LbpB–Lf complex forms a monodisperse stable complex (*Figure 2—figure supplement 1*). Therefore, crystallization screening focussed on both *Nm* and *Ng*LbpB–Lf complexes using commercial broad-matrix screens. Initial lead conditions were optimized, however, diffraction quality crystals could only be grown of the *Nm*LbpB–Lf complex. Data were collected at the GM/CA ID-D beamline at the Advanced Photon Source and the structure of the *Nm*LbpB–Lf complex solved to 2.85 Å resolution using molecular replacement in space group P4$_3$2$_1$2. Model building was performed in COOT and the structure refined using phenix.refine to final $R_{work}/R_{free}$ values of 0.20/0.25 (*Figure 3A* and *Supplementary file 1b*). One complex was observed in the structure containing one molecule of Lf (consisting of ordered N- and C-lobes) and one molecule of *Nm*LbpB (consisting of an ordered N-lobe, but partially ordered C-lobe).

The overall structure of the *Nm*LbpB–Lf complex has an overall L-shape (*Figure 3B*). The N-lobe of *Nm*LbpB is composed of residues 1–342 forming two domains, an N-terminal handle domain (residues 45–173) and an eight-stranded β-barrel domain (residues 174–342). The handle domain contains four anti-parallel β-strands and an α-helix between β-strand 1 and 2. The β-barrel domain is packed against the β-sheet of handle domain. The C-lobe of *Nm*LbpB is composed of residues 359–718 and found largely disordered in the crystal structure as evident from higher *B*-factors for this region compared to the rest of the structure (*Figure 3C*). The C-lobe also contains a handle domain (residues 359–540) and an eight-stranded β-barrel domain (residues 541–718). While the C-lobe β-barrel domain shares homology with the N-lobe β-barrel domain, the C-lobe handle domain is different as it is composed of a six-stranded β-sheet packed against the β-barrel domain and flanked by two antiparallel β-strands. Importantly, the C-lobe of LbpB contains several long anionic loops, however, none of these were observed in the electron density likely due to their flexibility.

The Lf structure also consists of N- and C-lobes with each lobe further divided into two subdomains termed N1 and N2 (N-lobe) and C1 and C2 (C-lobe). In the presence of iron, the subdomains of each lobe are bridged together forming the closed state (*Baker and Baker, 2005*; *Sill et al., 2016*). However, in absence of iron, these lobes are known to undergo conformational changes to adopt an open conformation (*Baker and Baker, 2005*; *Sill et al., 2016*). Structural alignment of Lf alone (PDB ID 2BJJ) with Lf in our complex (RMSD 1.3 Å) shows that both lobes are in closed conformations, as expected since we observe an iron in each lobe, and that Lf undergoes minimal changes upon binding

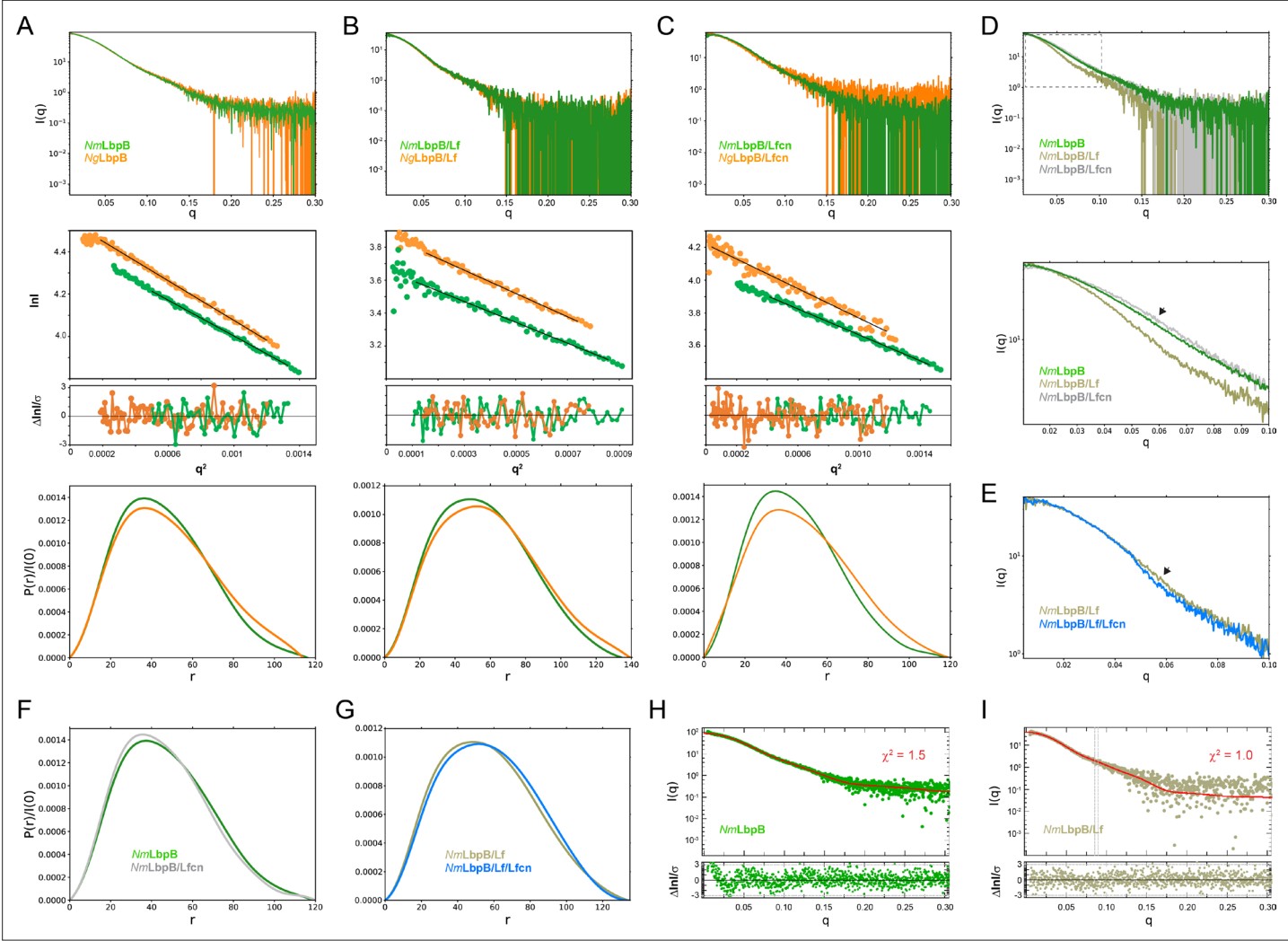

**Figure 2.** Solution structures of *Nm*/*N. gonorrhoeae* LbpB (*Ng*LbpB) alone and in complexes using size-exclusion chromatography small-angle X-ray scattering (SEC-SAXS). To compare the overall structures between the two species, SEC-SAXS analysis for both *N. meningitidis* LbpB (*Nm*LbpB) (green) and *Ng*LbpB (orange) was performed with resulting scattering profiles (top), Guinier plots (middle, offset for clarity), and P(r) plots (bottom) with closely matching $R_g$ and $D_{max}$ values alone (**A**), in complex with lactoferrin (Lf) (**B**), and in complex with lactoferricin (Lfcn) (**C**). Since minimal differences were observed between the *Nm* and *Ng* samples, this supports the conclusion that LbpB has a conserved overall structure. (**D**) A superposition of scattering profiles for *Nm*LbpB alone (green), in complex with Lf (olive), and in complex with Lfcn (grey). The bottom panel shows a zoomed view at lower *q* values to highlight the large change for the Lf complex and the small, but reproducible change observed for the Lfcn complex (green vs grey). Here, a significant difference in the shape of the curve was observed with Lf binding (green vs olive) confirming complex formation. And while small, a change upon Lfcn binding (green vs grey) could also be observed to confirm an overall structure change, albeit, much smaller given that the ligand is a peptide. (**E**) A zoomed view at lower *q* values comparing the *Nm*LbpB–Lf complex in the absence (olive) and presence (blue) of Lfcn. Again, a reproducible small change is observed in the presence of Lfcn, confirming binding of the peptide. (**F**) P(r) plots for *Nm*LbpB in the absence (green) and presence of Lfcn (grey) show that small structural changes occur upon Lfcn binding. (**G**) P(r) plots for *Nm*LbpB–Lf in the absence (olive) and presence of Lfcn (blue), similarly showing that small structural changes occur upon Lfcn binding even in the presence of Lf. (**H**) A Crysol plot of the calculated scattering curve for the *Nm*LbpB structure (red line) with the experimental scattering profile (green). (**I**) A Crysol plot of the calculated scattering curve for the *Nm*LbpB–Lf complex structure (red line) with the experimental scattering profile (olive). Panels H and I provide confirmation that our X-ray crystal structure of *Nm*LbpB–Lf agrees well with the 'in-solution' structures of *Nm*LbpB alone and in complex Lf.

The online version of this article includes the following figure supplement(s) for figure 2:

**Figure supplement 1.** Static small-angle X-ray scattering (SAXS) analysis of lactoferrin-binding protein B (LbpB) alone and in complex with lactoferrin.

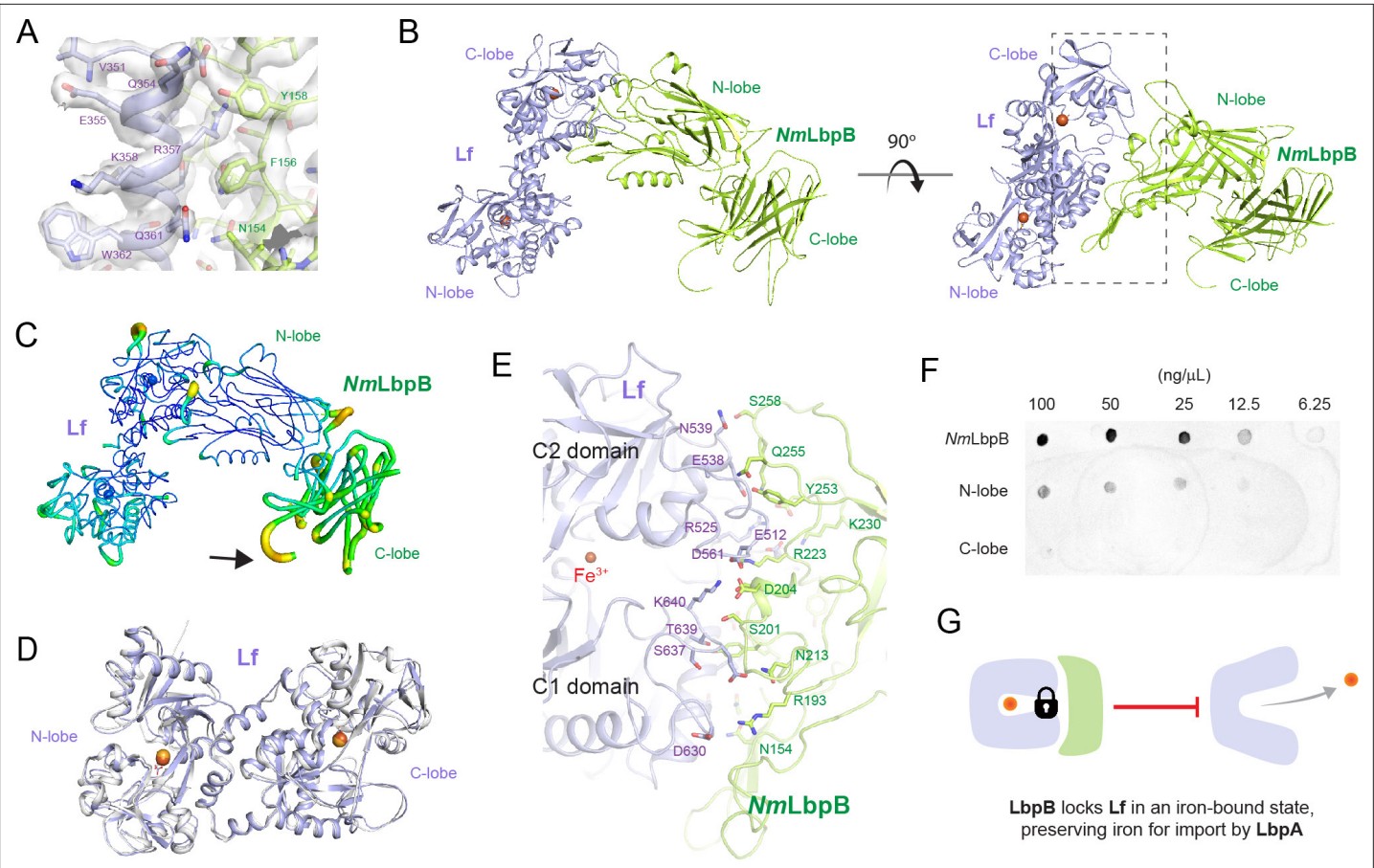

**Figure 3.** The 2.85 Å crystal structure of *N. meningitidis* LbpB (*Nm*LbpB) in complex with human lactoferrin. (**A**) Zoomed view at the interface between *Nm*LbpB and lactoferrin (Lf) depicting the quality of the electron density shown as a grey isosurface (2F$_O$–F$_C$, 1.0$\sigma$). (**B**) Orthogonal views of the complex with *Nm*LbpB in green, Lf in violet, and the iron atoms as red spheres. The N-lobe of *Nm*LbpB interacts with only the C-lobe of Lf along an extended interface. (**C**) The C-lobe of *Nm*LbpB has high *B*-factors with the large loops of this lobe not observed in our structure; the black arrow indicates the putative location of these loops. (**D**) An alignment of Lf from the complex with the structure of uncomplexed Lf (PDB ID 2BJJ) shows very little conformational changes upon binding *Nm*LbpB (root-mean-square deviation (RMSD) of 1.3 Å). (**E**) A zoomed view of the binding interface shows extensive interactions along an elongated surface covering both the C1 and C2 domains of Lf (buried surface area 1760.8 Å²). (**F**) Solid-phase-binding assays show Lf binds both full-length and N-lobe *Nm*LbpB, but not C-lobe only, supporting the observations in the complex structure. (**G**) Much like what has been proposed for the role of transferrin-binding protein (Tbp) B in the Tbp system, here we propose that lactoferrin-binding protein B (LbpB) also serves to bind and lock Lf in an iron-bound state for delivery to LbpA for iron import.

The online version of this article includes the following source data for figure 3:

**Source data 1.** The 2.85 Å crystal structure of *N. meningitidis* LbpB (*Nm*LbpB) in complex with human lactoferrin.

*Nm*LbpB (*Figure 3D*). Further, the calculated SAXS scattering curve from our crystal structure agrees well with our experimental SAXS scattering curve in solution (*Figure 2H1*).

The interaction of *Nm*LbpB with Lf is mediated exclusively through the N-lobe of *Nm*LbpB with the C-lobe of Lf (*Figure 3B*). This interface consists of regions along both the C1 and C2 domains of Lf including residues D630, S637, T639, K640 (C1 domain) and E512, R525, E538, N539, D561 (C2 domain) (*Figure 3B and E*). Interacting simultaneously with the C1 and C2 domains of Lf, the N-lobe of *Nm*LbpB has an extended interaction interface that includes residues N154, R193, S201, D204, N213, R223, K230, Y253, Q255, and S258 (*Figure 3B and E*). Given our crystal structure indicated that binding to Lf was exclusively along the N-lobe of *Nm*LbpB, we used solid-phase-binding assays to show that indeed Lf could only interact with the full length and N-lobe only constructs, but not with the C-lobe only (*Figure 3F and Figure 3—source data 1f*).

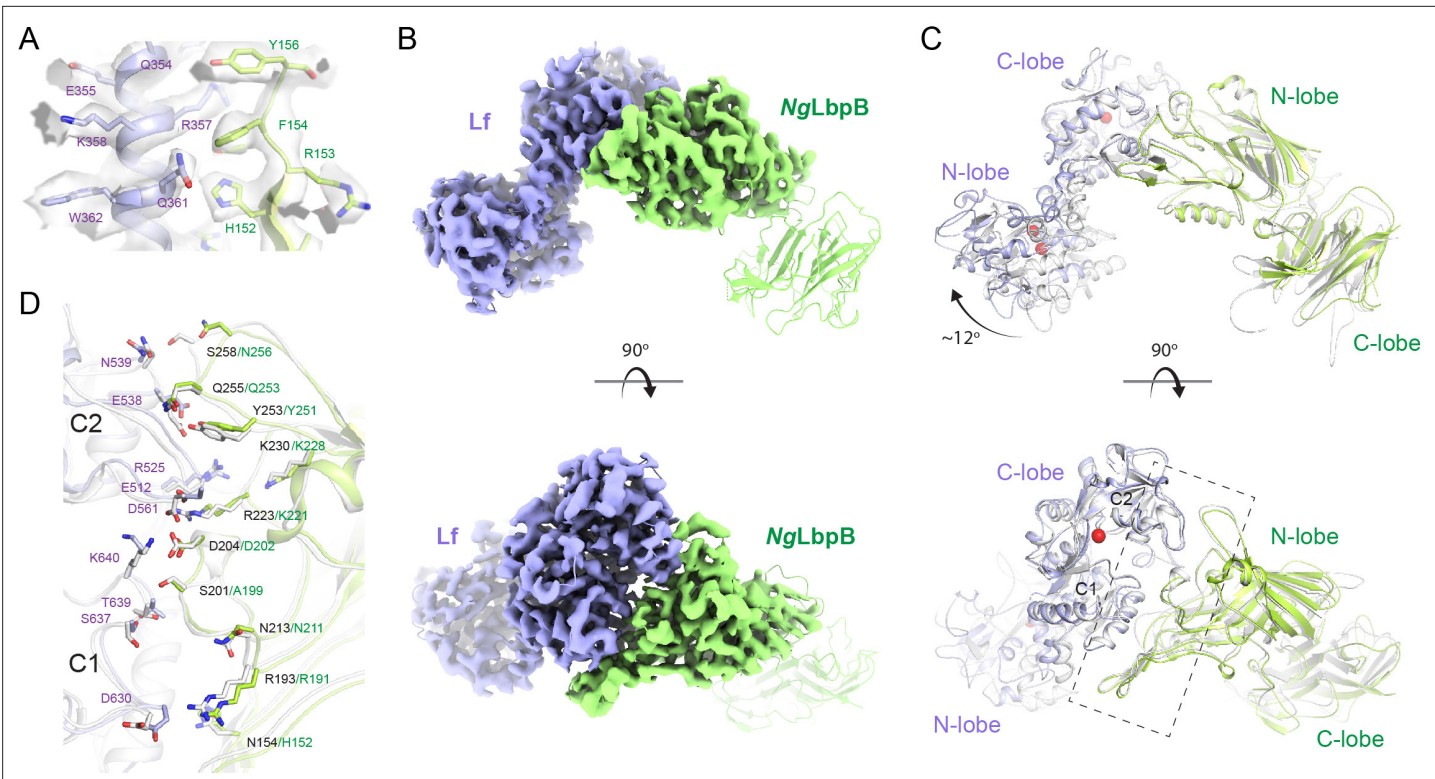

**Figure 4.** The 3.65 Å cryoEM structure of *N. gonorrhoeae* LbpB (*Ng*LbpB) in complex with human lactoferrin. (**A**) Zoomed view at the interface between *Ng*LbpB and lactoferrin (Lf) depicting the quality of the density shown as a grey isosurface. (**B**) Orthogonal views of the full cryoEM map with *Ng*LbpB in green and Lf in violet. (**C**) Orthogonal view of an alignment of the *Ng*LbpB–Lf cryoEM structure (green/violet) with the *N. meningitidis* LbpB (*Nm*LbpB)–Lf crystal structure (grey) (RMSD 1.4 Å along the interacting domains). (**D**) A zoomed view of the binding interface shows extensive interactions along an elongated surface covering both the C1 and C2 domains of Lf (buried surface area 1604 Å$^2$).

The online version of this article includes the following source data and figure supplement(s) for figure 4:

**Figure supplement 1.** CryoEM data processing workflow for *N. gonorrhoeae* LbpB (*Ng*LbpB)–Lf complex.

**Figure supplement 2.** Lactoferrin binding to *N. gonorrhoeae* LbpB (*Ng*LbpB) and *Ng*LbpB mutants.

**Figure supplement 2—source data 1.** Lactoferrin binding to *N. gonorrhoeae* LbpB (*Ng*LbpB) and *Ng*LbpB mutants.

**Figure supplement 3.** Structural comparison of *N. meningitidis* LbpB (*Nm*LbpB) to *Nm*TbpB.

## The cryoEM structure of *Ng*LbpB in complex with human Lf

Crystallization with *Ng*LbpB was unsuccessful, therefore, we used cryoEM to determine the structure of *Ng*LbpB in complex with Lf to 3.65 Å resolution (*Supplementary file 1c*). Very good density was observed for side chains of both lobes of Lf and for the N-lobe of *Ng*LbpB allowing unambiguous model building and refinement, however, almost no density was observed for the C-lobe of *Ng*LbpB, unless map contours were reduced drastically (*Figure 4A and B* and *Figure 4—figure supplement 1*). An alignment of the *Nm* and *Ng*LbpB complexes with Lf had an overall RMSD of 3.7 Å, however, an improved RMSD of 1.4 Å when aligning just along the interacting domains (C-lobe of Lf and N-lobe of LbpB), indicating a conserved binding interface between *Nm* and *Ng* (*Figure 4C and D*). Additionally, the N-lobe of Lf in the *Ng*LbpB–Lf cryoEM structure is positioned ~12° relatively to the *Nm*LbpB–Lf crystal structure. Whether this structural difference in the N-lobe of Lf is just an artifact of crystal packing or if it is truly important for function in LbpB remains to be determined.

## Mutagenesis studies targeting the binding interface between LbpB and Lf

In the *Nm/Ng*LbpB–Lf complex structures, LbpB interacts with both the C1 and C2 subdomains within the C-lobe of Lf in an extended interface (*Figures 3B, E ,, 4C and D*). In the *Nm*LbpB–Lf complex, the interaction interface is stabilized by extensive hydrogen bonding and salt bridge interactions

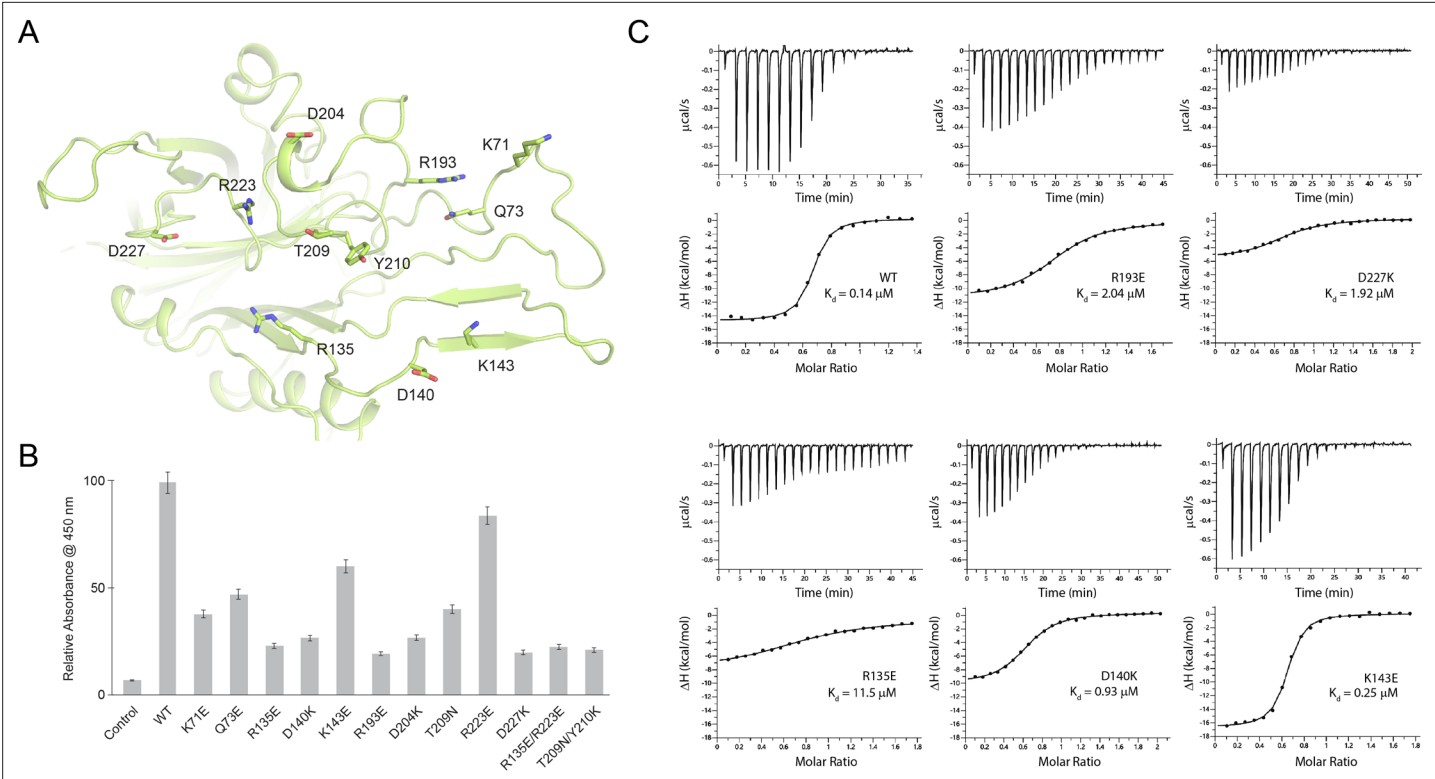

**Figure 5.** Probing the binding interface of *N. meningitidis* LbpB (*Nm*LbpB) with lactoferrin. (**A**) A zoomed view of the N-lobe of *Nm*LbpB along the lactoferrin (Lf) interaction interface, highlighting primary residues involved in binding. (**B**) Enzyme-linked immunosorbent assays (ELISAs) to test the effects of structure-guided mutations of *Nm*LbpB on Lf binding along the interaction interface. (**C**) Analysis of the binding parameters using isothermal titration calorimetry analysis of wild type and mutants of *Nm*LbpB measuring the effects on Lf binding.

encompassing a buried surface area of ~1760.8 Å$^2$ (*Supplementary file 1d*). Based on analysis of our crystal structure, we performed mutagenesis of interface residues in *Nm*LbpB important for mediating Lf binding (*Figure 5*). Those included K71, Q73, R135, K140, K143, R193, D204, T209, Y210, R223, and D227, most of which are conserved in *Ng*LbpB (*Figure 4D* and *Supplementary file 1e*). These mutants all expressed similar to wild type and could be purified using the same method, indicating they were stably folded. The effect of these mutations on Lf binding was first tested with solid-phasing binding assays (*Figure 4—figure supplement 2* and *Figure 4—figure supplement 2*; *Source data 1*) and with enzyme-linked immunosorbent assays (ELISAs) and compared to wild type (*Figure 5B*). The results showed that all of the mutations had a significant effect on Lf binding with most resulting in more than ~50 % reduction in binding except for mutations Q73E, K143E, and R223E. Those having at least a ~75 % reduction in binding included R135E, D140K, R193E, D204K, D227K, R135E/R223E, and T209N/Y210K. To further characterize Lf binding to the mutants, we performed isothermal titration calorimetry (ITC) analysis using a MicroCal iTC200 to determine thermodynamics parameters of binding, including $K_d$ values, and compared the results of the mutants to wild type (*Figure 5C* and *Supplementary file 1f*). Representative ITC plots show a $K_d$ value of 0.14 µM for Lf binding to wild-type *Nm*LbpB with a $\Delta H$ of −15.8 kcal/mol and an *n* value of ~1. Agreeing well with the ELISA analysis, $\Delta H$ values for the all the mutants except for K143E were significantly reduced ranging from −5.4 to −11.8 kcal/mol (R135E, D140K, R193E, and D227K) and accompanied by increases in $K_d$ values ranging from 9- to 115-fold over wild type.

## Mutagenesis studies targeting the putative binding interface between LbpB and Lfcn

To gain more insight into the binding of Lfcn with LbpB, we performed SEC-SAXS experiments with and without Lfcn, finding that the presence of Lfcn produced a small, but reproducible, change in the scattering profile at lower *q* values in the range of 0.04–0.08 (*Figure 2C and D*). A similar change

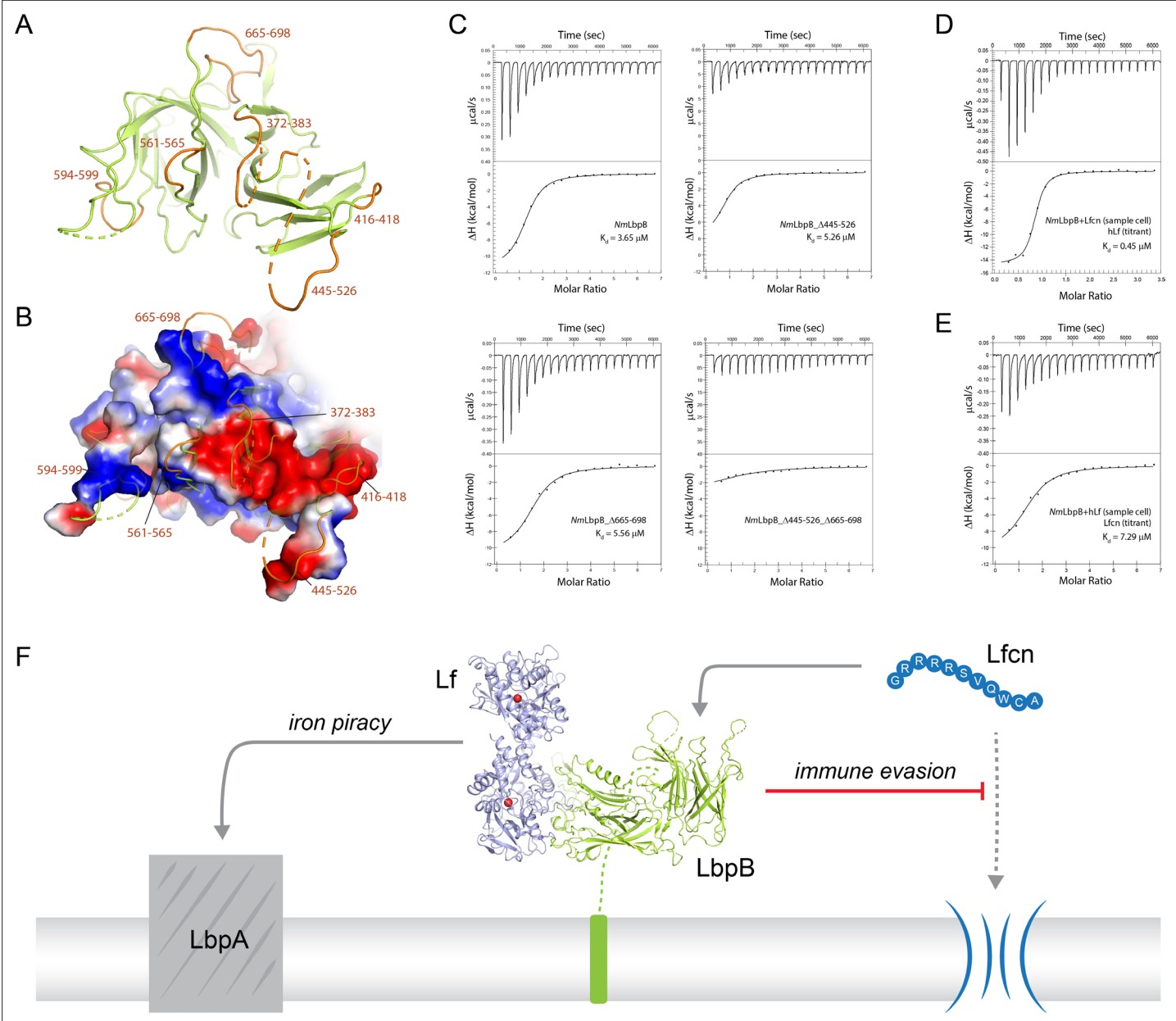

**Figure 6.** Probing the putative binding interface of *N. meningitidis* LbpB (*Nm*LbpB) with lactoferricin. (**A**) A zoomed view of the C-lobe of *Nm*LbpB with the loops indicated in orange. (**B**) An electrostatic surface potential representation along the C-lobe of *Nm*LbpB depicting the charged surfaces, including a strongly electronegative region (red). (**C**) Analysis of the binding parameters using isothermal titration calorimetry (ITC) analysis of wild type and loop-deletion mutants of *Nm*LbpB measuring the effects on lactoferricin (Lfcn) binding. (**D**) ITC analysis of the *Nm*LbpB–Lfcn complex titrated with lactoferrin (Lf), showing comparable binding to *Nm*LbpB alone. (**E**) ITC analysis of the *Nm*LbpB–(Lf) complex titrated with Lfcn, showing comparable binding to *Nm*LbpB alone. (**F**) Model for the dual function of LbpB in mediating Neisserial pathogenesis by serving in both iron piracy and as an antimicrobial peptide (AMP) sink. While not shown, processing by NalP produces a soluble version of LbpB which can diffuse into the host environment to actively locate and neutralize AMP threats.

along lower *q* values in the range of 0.05–0.07 was also observed for the *Nm*LbpB–Lf complex when Lfcn was added (*Figure 2E*). Together, these data support the hypothesis that Lfcn binding is distinct from Lf binding, further supporting that Lfcn binding is along the C-lobe of LbpB. While we were able to co-crystallize Lfcn with *Nm*LbpB–Lf and solve the structure, the C-lobe of LbpB remained highly disordered and therefore, we were unable to determine the binding interactions. Therefore, to probe the putative binding site of Lfcn, we analyzed the *Nm*LbpB–Lf structure to determine the boundaries of the C-lobe loops (*Figure 6A*). An electrostatic surface representation of this region shows it to

be highly charged with at least one large strongly electronegative region along loops consisting of residues 372–383, 416–418, and 445–526; a more accurate analysis of the electrostatics here is not possible given the lack of order in the loops (*Figure 6B*).

Based on these loop boundaries, we made deletion mutants with each lacking one of the loops of the C-lobe of *Nm*LbpB. Albeit to varying levels, these mutants all expressed and could be purified using the same method as for wild type, indicating they were stably folded. We then analyzed the effects of these deletions on Lfcn binding in comparison to wild type using ITC analysis on a Nano ITC instrument (*Figure 6C* and SI Table S7). Our results show that Lfcn binds to wild-type *Nm*LbpB with a $K_d$ of 3.7 µM, a $\Delta H$ of −11.4 kcal/mol, and an $n$ value of ~1. The loop deletions Δ372–383, Δ416–418, Δ561–565, Δ594–599, and Δ665–698, however, showed only minimal differences in $\Delta H$ and no significant change in $K_d$, except for Δ445–526 ($\Delta H$ of −9.6 kcal/mol and a ~2.5 -fold increase in $K_d$). All $n$ values here are larger than 1 and roughly twofold higher than that observed with Lf ($n$ value slightly below 1), an observation we attribute to inaccuracies in using absorbance measurements at $\lambda_{280}$ for determining protein and peptide concentrations. Loops containing residues 445–526 and 665–698 contain charged patches that have been shown to affect Lfcn binding (*Morgenthau et al., 2014a*; *Ostan et al., 2017*), however, individually removing them did not appear to abrogate binding in our study. However, a double loop-deletion mutant Δ445–526_Δ665–698 showed a significantly reduced $\Delta H$ with only barely detectable binding. This confirms that these two loops together contribute to Lfcn binding in LbpB, which is likely mediated by the charged residues they contain. Exactly how Lfcn binding is mediated though by these loops will require additional structural studies.

Our SEC-SAXS experiments suggested that Lf and Lfcn had distinct, non-competing, binding sites on LbpB. Therefore, we next performed ITC experiments using a Nano ITC instrument to determine the binding properties of Lf to *Nm*LbpB in the absence or presence of Lfcn. Our results show that Lf binds to *Nm*LbpB alone with a $K_d$ of 0.45 µM and a $\Delta H$ of −15.1 kcal/mol (SI Table S8). Nearly identical binding parameters are observed for Lf binding to the preformed *Nm*LbpB–Lfcn complex, indicating that the presence of Lfcn has no effect on Lf binding (*Figure 6D* and SI Table S8). We next tested the effect of the presence of Lf on Lfcn binding. Our earlier results showed that Lfcn binds to *Nm*LbpB alone with a $K_d$ of 3.7 µM and a $\Delta H$ of −11.4 kcal/mol, while here we found that Lfcn binds to the *Nm*LbpB–Lf complex with a $K_d$ of 7.3 µM and a $\Delta H$ of −10.6 kcal/mol (*Figure 6E* and SI Table S8). Only ~ twofold difference in $K_d$ was observed here, however, no change in $\Delta H$ indicating that binding of Lfcn to *Nm*LbpB is not disrupted in the presence of Lf.

## Discussion

The role of LbpB in Neisserial pathogenesis has been debated over the years given that it appears to share many properties with TbpB, which is involved in iron scavenging from Tf, but has also been shown to be important in protecting against the host antimicrobial peptide Lfcn (*Morgenthau et al., 2014a*; *Morgenthau et al., 2014b*; *Roussel-Jazédé et al., 2010*). For its iron import role, studies have indicated that LbpB may bind multiple copies of Lf (*Ostan et al., 2017*). However, our studies here strongly support that LbpB binds Lf in a 1:1 complex, which is demonstrated by our SEC, SAXS, X-ray crystal structure, and cryoEM structure. Further, our ITC experiments also demonstrate 1:1 binding properties with an $n$ value (molar ratio) of ~1.

Our structures align well with the previously reported N-lobe only structure of LbpB (PDB ID 4U9C) with an RMSD of 0.34 Å (for 261 Cα atoms) compared to our *Nm*LbpB crystal structure. Additionally, *Nm*LbpB aligns well with *Nm*TbpB (PDB ID 3V8U) with an overall RMSD of 1.8 Å. The interacting lobes from LbpB–Lf and TbpB–Tf (PDB ID 3VE1) have an overall RMSD of 1.4 Å, further evidence that the Lbp and Tbp systems likely share a common mechanism in iron acquisition from their respective host iron-binding proteins (*Figure 4—figure supplement 3*; *Gray-Owen and Schryvers, 1996*; *Noinaj et al., 2013*; *Schryvers and Lee, 1989*). Here, surface anchored LbpB would selectively accumulate iron-bound Lf, preserving the bound iron for eventual import by LbpA (*Figures 3G and 6F*; *Calmettes et al., 2012*). This would ensure efficiency in iron scavenging for survival during pathogenesis since TbpA (and likely LbpA) can also bind with high affinity apo-Tf/Lf which lacks iron (*Krell et al., 2003*; *Noinaj et al., 2012b*; *Powell et al., 1998*).

Unlike TbpB, LbpB has also been implicated in mediating evasion of the host immune responses during infection by specifically neutralizing the antimicrobial peptide Lfcn (*Morgenthau et al., 2014b*; *Ostan et al., 2017*). This may occur at the surface of the pathogen or in solution following cleavage

and release by NalP (*Morgenthau et al., 2014a*; *Roussel-Jazédé et al., 2010*). Exactly how LbpB interacts with Lfcn has not been fully resolved, however, studies have pointed to several charged patches on the C-lobe as mediating this interaction (*Morgenthau et al., 2014a*; *Ostan et al., 2017*). Further, we report here that the loop containing the smaller charged region (residues 665–698) alone has little effect on Lfcn binding compared to wild type, yet does appear to synergize binding with the loop consisting of 445–526. Together, our structural studies, mutagenesis, and binding studies agree with the earlier reports and allow us to conclude that Lfcn specifically interacts with the C-lobe only of LbpB through primarily two loops consisting of residues 445–526 and 665–698. It has been shown that LbpB can protect against other peptides as well, however, whether binding of these other peptides involves the same loops or charged interfaces remains to be determined. The presence of Lf had little effect on the binding of Lfcn and vice versa, demonstrating that each have non-overlapping and non-competing binding sites, refuting the previous notion that both may share a common binding site (*Brooks et al., 2014*). While we were able to grow crystals in the presence of Lfcn, the loops along the putative binding interface lacked sufficient density to observe the peptide itself or its interactions with LbpB. Therefore, further studies will need to be done here in order to fully understand the atomic details of this interaction. Together, our studies support a model where LbpB serves a dual function in Neisserial pathogenesis, in both iron import under iron-limiting conditions and as an antimicrobial peptide sink for the evasion of host immune responses (*Figure 6F*).

## Materials and methods
### Cloning, expression, and purification
The codon optimized LbpB from *N. meningitidis* strain MC58 (*Nm*LbpB) was purchased from Bio Basic. For expression, a gene fragment encoding residues 20–737 was subcloned in expression vector pHIS-Parallel2 with N-terminal 6× His-tag and Tobacco Etch Virus (TEV) protease cleavage site. For large-scale expression, the expression vector was transformed into BL21(DE3) cells and protein expression performed at 24 °C upon induction with 0.5 mM isopropyl β-D-1-thiogalactopyranoside (IPTG) for 9–12 hr. Cells were harvested by centrifugation at 6000 rpm and stored at −80 °C until further use.

For purification, frozen pellets were thawed and resuspended in lysis buffer (20 mM Tris–HCl, pH 7.5, 150 mM NaCl, 5 mM β-mercaptoethanol [BME], 25 mM imidazole) supplemented with 10 µg/ml DNaseI and 0.5 mM phenylmethylsulfonyl fluoride. Cells were lysed by three passages through Emulsiflex C3 high pressure homogenizer (Avestin). The cellular debris were removed by centrifugation at 18,000 rpm for 30 min at 4 °C. The supernatant was collected and subjected to affinity purification using 5 mL Hi-Trap column containing Ni-NTA resin (Qiagen) on an AKTA Pure 25 L protein purification system (GE Healthcare). The column was washed with 10 column volume (CV) lysis buffer and protein was eluted using elution buffer (20 mM Tris–HCl pH 7.5, 150 mM NaCl, 5 mM BME, 250 mM imidazole). The eluted protein was subjected to TEV protease digestion to remove the 6× His-tag. The digested protein was cleaned up using a Ni-NTA affinity column and the flowthrough collected. The concentrated flowthrough was further purified using SEC on a Superdex 200 10/300 GL Increase (GE Healthcare) column using 20 mM Tris–HCl pH 7.5, 150 mM NaCl. The elution fractions were run on an SDS–PAGE gel to examine purity. The fractions containing *Nm*LbpB were pooled and concentrated for further experiments.

The DNA sequences for the individual lobes (N- and C-lobes) encoding residues 20–362 and 381–737, respectively, were amplified using full-length *Nm*LbpB as template. These fragments were ligated into pHIS-Parallel2 with N-terminal 6× His-tag and TEV protease cleavage site. Also, point mutants were created using wild-type *Nm*LbpB plasmid using standard site-directed mutagenesis methods. These constructs were expressed and purified using the similar approach as full-length *Nm*LbpB as described above.

The *Ng*LbpB gene fragment encoding residues 20–728 was amplified from the genomic DNA of *N. gonorrhoeae* strain FA19 with primers containing restriction sites for NcoI and XhoI. The gene fragment was cloned into pHIS-Parallel2 plasmid for expression. The construct was confirmed by DNA sequencing. The expression and purification of *Ng*LbpB were performed using the same approach as for full-length *Nm*LbpB as described above.

For complex formation, purified *Nm/Ng*LbpB was incubated with twofold molar excess of human holo-Lf (RayBiotech, Inc) at 4 °C for 1 hr. The complex was separated from excess Lf using SEC on a

Superdex 200 10/300 GL Increase column (GE Healthcare). Complex formation was confirmed using SDS–PAGE gel. The fractions containing Nm/NgLbpB–Lf were pooled and concentrated for further experiments.

## Dot blots

For dot blot assays with NmLbpB wild type and mutants, 10 µl of each sample was spotted on polyvinylidene difluoride (PVDF) membrane and air dried. The membrane was then blocked by incubation of the membrane in 5 % bovine serum albumin (BSA) for 1 hr followed by three washes with 1× PBST (1× phosphate buffered saline [PBS] + 0.005 % Tween-20) 5 min each. Potential ligand binding was performed by incubating the blots in 20 µg/ml holo-Lf in PBST +0.5 % BSA for 1 hr. Unbound ligand was removed by three washes with PBST. The blots were probed using horse radish peroxidase (HRP) enzyme-conjugated rabbit anti-human Lf antibody (RayBiotech, Inc) for 1 hr. Excess antibody was removed by washing using three washes of 1× PBST and 1× PBS each. Finally, enhanced chemiluminescence (ECL) substrate was used for the detection of HRP's enzymatic activity.

For dot blot assays with NgLbpB wild type and mutants, 10 µl of each sample was spotted on activated PVDF membrane and air dried. 1× PBS with BSA (5%) was used for blocking unoccupied sites followed by three washes with 1× PBST. The HRP-conjugated holo-Lf (Lf-HRP) probe was used to monitor Lf binding. Excess probe was removed by washes with 1× PBST and 1× PBS. Finally, ECL substrate was used for the detection of HRP activity.

All dot blot assays were performed at least in triplicate with representative data shown.

## Size-exclusion chromatography small-angle X-ray scattering

Purified samples were subjected to inline SEC-SAXS. The scattering data were collected at beamline 18-ID of Biophysics Collaborative Access Team (BioCAT) of the Advanced Photon Source, Argonne National Laboratory. The data were analyzed and final plots made using BioXTAS RAW (*Hopkins et al., 2017*) and ATSAS (*Manalastas-Cantos et al., 2021*). First, the data were reduced and the data range for scattering curves were selected. Upon averaging of the data, the q-range and molecular weight information were obtained by analyzing Guinier plot. The pair-distance distribution curves were calculated using GNOM. Theoretical scattering for the X-ray and cryoEM structures was calculated and compared with experimental scattering curves using Crysol.

## Crystallization, data collection, and structure determination using X-ray crystallography

For crystallization, NmLbpB–Lf was concentrated to 7.4 mg/ml. Initial crystallization trials were performed using commercial screens using hanging-drop vapour-diffusion method at 20 °C. Crystal hits identified from these screens were optimized. Diffraction quality crystals of NmLbpB–Lf were obtained in 2.0 M ammonium sulphate, 20 mM Tris, pH 8.5 at 20 °C. Crystals were harvested and diffraction data were collected at the GM/CA 23ID-D beamline at the Advanced Photon Source, Argonne National Laboratory using Pilatus 6 M detector at 100 K. The data were processed using HKL2000 (*Otwinowski and Minor, 1997*) in space group $P4_32_12$ with unit cell parameters $a = b =$ 120.39, $c = 207.38$, $\alpha = \beta = \gamma = 90.0$.

Initial phases were calculated by molecular replacement within PHASER (Phenix) (*Liebschner et al., 2019*; *McCoy et al., 2007*) using the structure of diferric human Lf (PDB ID: 2BJJ) and structure of the N-lobe of NmLbpB (PDB ID: 4U9C). Furthermore, the model was build using Coot (*Emsley et al., 2010*) and refined using phenix.refine (Phenix) (*Liebschner et al., 2019*). The final model was refined to $R_{work}$ and $R_{free}$ values 0.20 and 0.25, respectively, at 2.85 Å resolution. The geometric analysis was performed using Molprobity (Phenix) (*Chen et al., 2010*; *Liebschner et al., 2019*). Structural analysis and figure preparation were performed using PyMol (Schrödinger). Final figures were assembled using Adobe Photoshop and Illustrator.

## Grid preparation, data collection, and structure determination using cryoEM

Purified NgLbpB–Lf complex was applied to Quantifoil R 3.5/1 Cu 200 grids, that were first glow discharged using a Pelco EasiGlow instrument, and plunge-frozen using Vitrobot Mark IV (Thermo Fisher Scientific). Grids were screened and the grid with optimal particle distribution and ice-thickness

was used for data collection using a Titan Krios G1 microscope equipped with a Gatan K3 direct electron detector and a Gatan Quantum energy filter. Movies were collected in super-resolution counting mode (pixel size 0.54 Å) with a total dose of 53.68 e$^-$/Å$^2$. A total 4966 movies were collected and subjected to motion correction using MotionCor2 (*Zheng et al., 2017*). The subsequent data processing was performed using cryoSPARC (*Punjani et al., 2017*). CTF parameters were obtained using 'Patch CTF estimation (multi)'. An initial set of templates were made by first using 'Blob picker' followed by '2D Classification'. The best 2D classes were then fed into 'Template picker' and subjected to multiple rounds of 2D classification to remove junk particles. The best classes representing different particle orientation were then used for 'Ab-initio Reconstruction'. After two rounds of 'Ab-initio Reconstruction' followed by 'Heterogeneous refinement', the best-looking class was subjected to 'Non-uniform Refinement', which yielded a final reconstruction to 3.65 Å resolution. A model of the *Ng*LbpB–Lf structure was prepared based on the *Nm*LbpB–Lf crystal structure and placed and fit within the map using ChimeraX (*Pettersen et al., 2021*). The cryoEM map contained well-resolved density at the protein–protein interface, however, the C-lobe of *Ng*LbpB could only be placed as a rigid body given only minimal density even when map contours were reduced. The structure was refined using 'Real-space refinement' within Phenix (*Liebschner et al., 2019*) and figures and analysis performed using ChimeraX (*Pettersen et al., 2021*) and PyMOL (Schrödinger). Final figures were assembled using Adobe Photoshop and Illustrator.

## Enzyme-linked immunosorbent assays

For binding analysis with Lf, we performed sandwich ELISAs. The *Nm*LbpB variants were coated on hydrophobic plates by first incubating the plates with 0.1 mg/ml of each protein at 4 °C overnight. *Acinetobacter baumannii* BamB (a kind gift from Robert Stephenson) was included as a negative control. After incubation, wells were washed three times with 1× PBST and blocked with 5 % BSA for 30 min. Subsequently, BSA was removed and the wells were washed three times with 1× PBST. For ligand binding, the wells were incubated with the Lf for 30 min. Following that, plates were washed three times with 1× PBST and probed with HRP-conjugated rabbit anti-human Lf antibody (RayBiotech, Inc) for 30 min. The antibody was washed away with three washes of 1× PBST and 1× PBS. The assay was then developed using 1-Step Ultra TMB-ELISA substrate solution (Thermo Scientific). Finally, the assay was stopped upon addition of 0.18 M sulfuric acid. For quantitative data, the absorbance at 450 nm was measured using a Spectramax M2e microplate reader (Molecular Devices). All the experiments were performed in triplicate and repeated at least twice. Similarly, to study Lf binding of the *Ng*LbpB wild type and mutants, ELISAs were performed using HRP-conjugated holo-Lf (Lf-HRP) as a probe for Lf binding. *Pisum sativum* Toc75 POTRA (a kind gift from Karthik Srinivasan) was included as a negative control.

All ELISA experiments were performed in triplicate with standard errors shown.

## Isothermal titration calorimetry

To monitor the properties of Lf binding to the *Nm*LbpB variants, ITC experiments were performed using a MicroCal iTC200 ITC calorimeter (Malvern Panalytical). The *Nm*LbpB variants and Lf proteins were buffer exchanged into 1× PBS (PBS: 137 mM NaCl, 2.7 mM KCl, 10 mM Na$_2$HPO$_4$, 1.8 mM KH$_2$PO$_4$, pH 7.5) by SEC. The *Nm*LbpB variants and Lf were concentrated to 30 and 300 µM concentration, respectively. Lf was titrated into ITC sample cell containing the *Nm*LbpB variants at 25 °C. Data analysis was performed using Origin 7.0 (OriginLab).

Lfcn binding to wild type and loop-deletion mutants of *Nm*LbpB was measured using a Nano ITC instrument (TA instruments). The sample cell contained 30 µM of protein with 600 µM of Lfcn injected over 20 injections with stirring at 300 RPM at a temperature of 10 °C. The ITC data were analyzed using minimized independent binding model with the instrument NanoAnalyze software.

ITC experiments to study the effect of presence of Lfcn on Lf binding to *Nm*LbpB and vice versa, were performed using a Nano ITC instrument (TA instruments). The protein in the sample cell was loaded at 30 µM concentration with stirring at 300 rpm at 25 °C for Lf and at 10 °C for Lfcn titrations at concentrations of 300 and 600 µM, respectively. The ITC data were analyzed using a minimized independent binding model using the instrument NanoAnalyze software.

All ITC experiments were performed with at least two replicates with representative data shown.

## Acknowledgements

We thank the Noinaj Lab for advice and helpful discussions on these studies. We thank Dr. Thomas Klose (Purdue CryoEM Facility) and Runrun Wu (Noinaj Lab) for their assistance with collection of the cryoEM data. This work was supported by the Purdue Research Foundation (RY) and by the Department of Biological Sciences (NN). The cryoEM studies were funded, in part, with support from the Indiana Clinical and Translational Sciences Institute funded, in part by Grant Number UL1TR002529 from the NIH, National Center for Advancing Translational Sciences, Clinical and Translational Sciences Award. We also thank the beamline staffs of the BioCAT and GM/CA beamlines at the Advanced Photon Source (APS), Argonne National Laboratory for their help with data collection. GM/CA@APS has been funded by the National Cancer Institute (ACB-12002) and the National Institute of General Medical Sciences (NIGMS) (AGM-12006, P30GM138396). The BioCAT resources are supported by grant P30 GM138395 from the NIGMS of the NIH; use of the Pilatus 3 1 M detector was provided by grant 1S10OD018090 from NIGMS. This research used resources of the Advanced Photon Source, a U.S. Department of Energy (DOE) Office of Science User Facility operated for the DOE Office of Science by Argonne National Laboratory under Contract No. DE-AC02-06CH11357.

## Additional information

### Funding

| Funder | Grant reference number | Author |
|---|---|---|
| Purdue Research Foundation | | Nicholas Noinaj<br>Ravi Yadav |
| Department of Biological Sciences, Purdue University | | Nicholas Noinaj<br>Ravi Yadav |
| Indiana Clinical and Translational Sciences Institute | | Nicholas Noinaj |

The funders had no role in study design, data collection and interpretation, or the decision to submit the work for publication.

### Author contributions

Ravi Yadav, Conceptualization, Data curation, Formal analysis, Funding acquisition, Investigation, Methodology, Validation, Visualization, Writing – original draft, Writing – review and editing; Srinivas Govindan, Courtney Daczkowski, Investigation, Methodology; Andrew Mesecar, Resources; Srinivas Chakravarthy, Data curation, Formal analysis, Investigation, Methodology; Nicholas Noinaj, Conceptualization, Data curation, Formal analysis, Funding acquisition, Investigation, Methodology, Project administration, Resources, Supervision, Validation, Visualization, Writing – original draft, Writing – review and editing

### Author ORCIDs

Ravi Yadav  http://orcid.org/0000-0003-2879-0465
Srinivas Govindan  http://orcid.org/0000-0002-0414-7822
Nicholas Noinaj  http://orcid.org/0000-0001-6361-2336

### Decision letter and Author response

Decision letter https://doi.org/10.7554/eLife.71683.sa1
Author response https://doi.org/10.7554/eLife.71683.sa2

## Additional files

### Supplementary files

• Supplementary file 1. Supplementary Information.
 (a) Summary of size-exclusion chromatography small-angle X-ray scattering (SEC-SAXS) parameters.

Lactoferrin (Lf) has a calculated molecular weight of 76.3 kDa, lactoferricin (Lfcn) 1.4 kDa, *N. gonorrhoeae* LbpB (*Ng*LbpB) 78.4 kDa, and *N. meningitidis* LbpB (*Nm*LbpB) 79.5 kDa. (b) Data collection and refinement statistics for the *Nm*LbpB–Lf X-ray crystal structure. (c) Data collection and refinement statistics for the *Ng*LbpB–Lf cryoEM structure. (d) Summary of the intermolecular interactions between *Nm*LbpB and Lf. The information about interacting residues was obtained by QtPISA analysis. (e) Intermolecular interactions between *Ng*LbpB and Lf. The information about interacting residues was obtained by QtPISA analysis. (f) Summary of ITC parameters for lactoferrin binding to *Nm*LbpB mutants. These experiments were performed using a MicroCal iTC200 ITC calorimeter (Malvern Panalytical). (g) Summary of ITC parameters for lactoferricin binding to *Nm*LbpB loop deletions. These experiments were performed using a Nano ITC calorimeter (TA Instruments). (h) Summary of ITC parameters for lactoferrin and lactoferricin binding to *Nm*LbpB. These experiments were performed using a Nano ITC calorimeter (TA Instruments).

- Transparent reporting form
- Source data 1. Source data for figures.

## Data availability

All coordinates, structure factors, and cryo-EM maps have been uploaded to the PDB and/or the EMDB as follows: X-ray structure of NmLbpB in complex with human lactoferrin, PDB 7JRD; Cryo-EM structure of NgLbpB in complex with human lactoferrin, PDB 7N88 (EMD-24233).

The following dataset was generated:

| Author(s) | Year | Dataset title | Dataset URL | Database and Identifier |
|---|---|---|---|---|
| Yadav R, Noinaj N | 2021 | The crystal structure of lactoferrin binding protein B (LbpB) from Neisseria meningitidis in complex with human lactoferrin | https://www.rcsb.org/structure/7JRD | RCSB Protein Data Bank, 7JRD |
| Yadav R, Noinaj N | 2021 | The cryoEM structure of LbpB from N. gonorrhoeae in complex with lactoferrin | https://www.rcsb.org/structure/7N88 | RCSB Protein Data Bank, 7N88 |
| Yadav R, Noinaj N | 2021 | The cryoEM structure of LbpB from N. gonorrhoeae in complex with lactoferrin | https://www.ebi.ac.uk/pdbe/entry/emdb/EMD-24233 | Electron Microscopy Data Bank, EMD-24233 |

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
