## [Decision Letter]

**Acceptance summary:**

Neisseria infection in humans can lead to serious illnesses, including meningitis and gonorrhea. The study provides important structural details on the molecular interaction between LbpB, a receptor from pathogenic Neisseria bacteria, and the human iron-binding protein lactoferrin. An elegant combination of structural biology and biochemical techniques establishes the binding interaction between these proteins and hints at details of interactions between LbpB and lactoferricin, an antimicrobial peptide derived from lactoferrin itself. This work is especially important for the field of antimicrobial resistance.

**Decision letter after peer review:**

Thank you for submitting your article "Structural insight into the dual function of LbpB in mediating Neisserial pathogenesis" for consideration by *eLife*. Your article has been reviewed by 2 peer reviewers, and the evaluation has been overseen by a Reviewing Editor and Olga Boudker as the Senior Editor. The following individuals involved in review of your submission have agreed to reveal their identity: Matthew F. Barber (Reviewer #1); John Young (Reviewer #2).

1. Solving the LbpB-lactoferrin structure(s) provides an important advancement for the field that has been sorely missed – congrats to the authors on this great achievement!

2. Morgenthau et al., (PLOS One, 2014) previously demonstrated that removal of negatively charged sites in the LbpB C-lobe impairs protection against lactoferricin, and other previous studies had implicated the C-lobe of LbpB in lactoferricin recognition. In light of these past studies, it is unclear how our understanding of the LbpB-lactoferricin interaction is significantly advanced. From the title of the paper as well as the abstract, understanding the "dual" functions of LbpB seems to be a key goal of the work. Few of the loop mutations appear to have any effect on lactoferricin binding, and it is unclear how the double loop mutant affects the overall structure of the LbpB C-lobe. Any additional insights that could be gleaned regarding the LbpB-lactoferricin interaction would greatly strengthen the paper and support the general conclusions regarding the dual functions of LbpB.

3. All experiments need to have biological repeats to ensure reproducibility. It was not clear from the paper whether all experiments, particularly ITC experiments, have been repeated. Please, clarify and perform replicates as needed.

4. ITC experiments produced stoichiometry n-values that are about twofold higher for lactoferricin than lactoferrin. This surprising observation begs the question: is it possible that two lactoferricin peptides bind to LbpB?

5. Figure 2 is somewhat difficult to interpret. There are many individual panels, and it is difficult for a non-expert to understand what specific information is being conveyed in each figure panel. Please add more details in the figure legend and in the main text to guide the reader. The authors may also want to consider including this as supporting information rather than as the main figure.

6. Lines 145-148: The authors state: "the NmLbpB-Lf complex forms a monodisperse stable complex (data not shown)". It seems this is a rather important point in the context of this manuscript – the authors may want to include this data.

7. Figure 5: The (δ) H vs. Molar ratio plots for the D227K and R135E mutants look almost linear. It seems difficult to derive a kD value from such a plot. The authors should mention this as a caveat in the Results section.*Reviewer #1:*

Pathogenic Neisseria cause bacterial meningitis and gonorrhea in humans and are a major public health concern. The acquisition of essential nutrient iron and resistance to host-derived antimicrobial peptides are key factors in the ability of bacterial pathogens to colonize and cause disease. Previous work has demonstrated that lactoferrin binding protein B (LbpB) is a conserved surface lipoprotein in Neisseria that binds the abundant host iron-binding protein, lactoferrin. While a role for LbpB has been demonstrated in bacterial iron acquisition, the molecular details of this interaction have remained unclear for decades. Moreover, LbpB has previously been reported to mediate bacterial protection against lactoferricin, an antimicrobial peptide cleaved from lactoferrin itself. The details of the LbpB-lactoferricin interaction have been even more contentious over the years.

In this study, Yadav et al., use a combination of structural and biochemical approaches to determine the structure of the LbpB-lactoferrin complex. Mutagenesis experiments further support the molecular basis for this binding interface between the LbpB N-lobe and the lactoferrin C-lobe. These finding provide an important advancement in our understanding of the LbpB-lactoferrin interaction and will greatly aid in further studies to combat pathogenic Neisseria. The combination of structural approaches (crystallography and cryo-EM) along with other supporting experiments strongly support the authors conclusions regarding LbpB-lactoferrin interactions.

The authors also perform a series of experiments to probe the LbpB-lactoferricin interaction, providing some evidence supporting the existing model that lactoferricin interacts with the C-lobe of LbpB, at a distinct surface from lactoferrin. While the findings are consistent with previous work, much still remains to be determined regarding this second reported function of LbpB and its importance during Nesseria pathogenesis.*Reviewer #2:*

Yadav et al., utilize X-ray crystallography and cryogenic electron microscopy to determine high-resolution structures of the iron-scavenging lipoprotein LbpB from two different species of pathogenic bacteria (N. meningitidis and N. gonorrhoeae) bound to the human iron-binding protein Lactoferrin. Their data reveal the structures of LbpB are very similar between the two species and show that LbpB consists of two distinct lobes: a well-ordered N-terminal lobe which binds Lactoferrin, and a partially disordered C-terminal lobe. Using site-directed mutagenesis, the authors identify several charged residues on LbpB at the interaction interface which appear especially critical for mediating lactoferrin binding. They quantify the effects of these mutations on LbpB-lactoferrin binding using Isothermal Titration Calorimetry (ITC).

The authors also investigate the role of LbpB in protecting Neisseria against the cationic antimicrobial peptide lactoferricin. They show that the C-terminal lobe of LbpB interacts with lactoferricin and identify several flexible loop regions of LbpB which mediate this interaction. Furthermore, they quantify the effects of mutating these loop regions on lactoferricin binding using ITC. Since these loop regions were not well-resolved in their structures, the authors were unable to identify specific residue(s) which may mediate the LbpB-lactoferricin interaction. The authors state that obtaining high-resolution structural data in this C-terminal region will be the subject of future work.

The manuscript is well written, and the experiments are performed to a high technical standard. The experimental data fully support the authors' conclusions. Furthermore, this work has relevance for human health; Neisseria infection is a growing public health concern, particularly in light of the recent rise of antibiotic-resistant strains. As the authors point out, the Lbp system appears to be a promising target for new therapeutics.

---

## [Author Response]

The reviewers have discussed their reviews with one another, and the Reviewing Editor has drafted this to help you prepare a revised submission.1. Solving the LbpB-lactoferrin structure(s) provides an important advancement for the field that has been sorely missed – congrats to the authors on this great achievement!

We thank the reviewers for the positive comment and glad the manuscript has been well received.

2. Morgenthau et al., (PLOS One, 2014) previously demonstrated that removal of negatively charged sites in the LbpB C-lobe impairs protection against lactoferricin, and other previous studies had implicated the C-lobe of LbpB in lactoferricin recognition. In light of these past studies, it is unclear how our understanding of the LbpB-lactoferricin interaction is significantly advanced. From the title of the paper as well as the abstract, understanding the "dual" functions of LbpB seems to be a key goal of the work. Few of the loop mutations appear to have any effect on lactoferricin binding, and it is unclear how the double loop mutant affects the overall structure of the LbpB C-lobe. Any additional insights that could be gleaned regarding the LbpB-lactoferricin interaction would greatly strengthen the paper and support the general conclusions regarding the dual functions of LbpB.

The negatively charged loops are disordered in our crystal structure. Therefore, to further probe the binding region for lactoferricin peptide more than what has already been published, we used our structure to create the loop truncation mutants. This has not been performed previously and therefore, we tested whether other loops could be contributing to binding. Overall, our ITC binding results are consistent with the killing assays performed by Morgenthau et al., (PLOS One, 2014). In this previous study, only a single and double mutant were made to remove the charged regions, which made it unclear if both charged loop regions were required or just one. Here, we test both individually and together, finding that it appears that both are required for efficient lactoferricin binding. We have updated the manuscript to highlight this point further. And while we did observe some differences in expression levels between the loop mutants, we could purify them all using the same methods as with the wild type, suggesting their structures were stable.

3. All experiments need to have biological repeats to ensure reproducibility. It was not clear from the paper whether all experiments, particularly ITC experiments, have been repeated. Please, clarify and perform replicates as needed.

All the experiments were performed with replicates. ELISA and dot-blot assays were repeated at least in triplicate, whereas ITC experiments were performed at least twice. This has been updated in the Methods section accordingly.

4. ITC experiments produced stoichiometry n-values that are about twofold higher for lactoferricin than lactoferrin. This surprising observation begs the question: is it possible that two lactoferricin peptides bind to LbpB?

We thank the reviewer for the comment and we also discussed the n-values for lactoferrin and lactoferricin binding on our end. Since n-values for ITC are known to be dependent on exact concentrations, we rationalized that the observed n-values likely reflect inaccuracies in determining concentrations. We used A280 measurements for determining the concentrations of the proteins and peptide; a method that can have associated errors. While we cannot exclude that two lactoferricin peptides possibly bind to LbpB, we do not observe any obvious biphasic behavior in our ITC data. A structure of the peptide bound to LbpB is needed to fully resolve this interaction.

5. Figure 2 is somewhat difficult to interpret. There are many individual panels, and it is difficult for a non-expert to understand what specific information is being conveyed in each figure panel. Please add more details in the figure legend and in the main text to guide the reader. The authors may also want to consider including this as supporting information rather than as the main figure.

We thank the reviewer for the comment and agree this is a busy figure. However, we decided that having it compiled as presented was more efficient, rather than having many supplementary panels. As recommended, we have expanded the figure legend and the text to assist the reader understand the SAXS data more clearly.

6. Lines 145-148: The authors state: "the NmLbpB-Lf complex forms a monodisperse stable complex (data not shown)". It seems this is a rather important point in the context of this manuscript – the authors may want to include this data.

Since we already presented a significant amount of SAXS data, we left this out. However, as recommended, we have now included an additional supplementary figure with the details of the static SAXS data. We agree that it could assist others in possibly using static SAXS (as opposed to SEC-SAXS) to determine conditions which may improve their chances of growing well-ordered crystals.

7. Figure 5: The (δ) H vs. Molar ratio plots for the D227K and R135E mutants look almost linear. It seems difficult to derive a kD value from such a plot. The authors should mention this as a caveat in the Results section.

We thank the reviewers for the comment. Earlier drafts of the manuscript indeed had versions of the panels that were not scaled together, so it was easier to see the fits of the curves. We decided to scale them all to easier cross compare results for the mutants to the controls. To assure the reviewers that the fit is reasonable for these two mutants, we have included non-scaled versions, Author response image 1, which verifies that the data analyses has been reliably performed within the NanoAnalyze software. We have not included this in the revised version of the manuscript, however, we are happy to do so if the reviewers feel strongly that it should be done.

**Author response image 1. sa2fig1:**